# Iodine Immobilized UiO-66-NH_2_ Metal-Organic Framework as an Effective Antibacterial Additive for Poly(ε-caprolactone)

**DOI:** 10.3390/polym14020283

**Published:** 2022-01-11

**Authors:** Wei Chen, Ping Zhu, Yating Chen, Yage Liu, Liping Du, Chunsheng Wu

**Affiliations:** Institute of Medical Engineering, School of Basic Medical Sciences, Health Science Center, Department of Biophysics, Xi’an Jiaotong University, Xi’an 710061, China; weiwcchen@xjtu.edu.cn (W.C.); jewel121@stu.xjtu.edu.cn (P.Z.); ytc20201011@stu.xjtu.edu.cn (Y.C.); yageliu@xjtu.edu.cn (Y.L.)

**Keywords:** UiO-66-NH_2_ metal-organic framework, iodine, poly(ε-caprolactone), antibacterial

## Abstract

Iodine has been widely used as an effective disinfectant with broad-spectrum antimicrobial potency. However, the application of iodine in an antibacterial polymer remains challenging due to its volatile nature and poor solubility. Herein, iodine immobilized UiO-66-NH_2_ metal-organic framework (MOF) (UiO66@I_2_) with a high loading capacity was synthesized and used as an effective antibacterial additive for poly(ε-caprolactone) (PCL). An orthogonal design approach was used to achieve the optimal experiments’ conditions in iodine adsorption. UiO66@I_2_ nanoparticles were added to the PCL matrix under ultrasonic vibration and evaporated the solvent to get a polymer membrane. The composites were characterized by SEM, XRD, FTIR, and static contact angle analysis. UiO-66-NH_2_ nanoparticles have a high iodine loading capacity, up to 18 wt.%. The concentration of iodine is the most important factor in iodine adsorption. Adding 0.5 wt.% or 1.0 wt.% (equivalent iodine content) of UiO66@I_2_ to the PCL matrix had no influence on the structure of PCL but reduces the static water angle. The PCL composites showed strong antibacterial activities against *Staphylococcus aureus* and *Escherichia coli*. In contrast, the same content of free iodine/PCL composites had no antibacterial activity. The difference in the antibacterial performance was due to the different iodine contents in the polymer composites. It was found that MOF nanoparticles could retain most of the iodine during the sample preparation and storage, while there was few iodine left in the free iodine/PCL composites. This study offers a common and simple way to immobilize iodine and prepare antibacterial polymers with low antiseptic content that would reduce the influence of an additive on polymers’ physical properties.

## 1. Introduction

Metal-organic frameworks (MOFs), or coordination polymers, are a novel type of highly porous materials with a crystalline structure [1]. The flexible network, tunable pore sizes, and rich physicochemical properties enable them to be promising materials for gas separation, capture and storage, catalysis, chemical sensors, and biomedicine [2,3,4,5]. MOFs have ultrahigh porosity (up to 90% free volume), an enormous internal surface area, and surface area (beyond 6000 m^2^/g^3^), supporting their applications in molecules’ adsorption, drug loading, and release [4,6,7]. The incorporation of MOFs into a polymer as mixed-matrix membranes provides a solution to manipulate and process the crystalline and robust MOFs [8,9]. The composite combines the molecular sieving effect of MOFs but overcomes the brittleness, accelerating the molecular separation and industrial application. Additionally, in separation fields, MOFs/polymer composite also could be used as an antibacterial material. UiO-66 and MOF-525 nanoparticles disperse uniformly in poly(ε-caprolactone) (PCL) even in a high loading capacity [10]. The mixed-matrix membranes exhibit the integrity of the pore structure of UiO-66 through dye separation. The MOF-525/PCL membrane shows effective antibacterial activity against *Escherichia coli* by the generation of reactive oxygen species. Additionally, the open Zr site of UiO-66 improved the adhesion on the interface of the PCL matrix.

Iodine is a well-known antimicrobial agent against aerobic and anaerobic bacteria as well as viruses, chlamydia, and fungi [11,12,13]. Iodine is slightly soluble in water and the solubility is enhanced in the presence of iodine ions. The free iodine is a toxic, irritating, and physiologically active molecule, limiting its biomedicine applications. One solution is anchoring iodine to a molecule or polymer-forming complexes [14]. The most extensively used polymer-iodine complex is the polyvinylpyrrolidone-iodine (PVP-I) complex. However, the PVP-I complex has some shortages such as causing allergic reactions, staining tissues, and being releasing in an uncontrolled manner. Loading of iodine with nanomaterials, such as graphene, halloysite, ZIF-8, CD-MOF, etc., is a convenient way to control the loading and release behavior [15,16,17]. A cross-linked cyclodextrin metal-organic framework (CD-MOF) is used for controlled iodine loading and release [18]. A slow and sustained release of iodine is found in artificial saliva. When added to the hydroxyethyl cellulose gel, the sustained release could extend up to 5 days, slower than CD-MOF@I_2_. Meanwhile, the PVP-I complex consists of water-soluble polymers, which is undesirable in the preparation of a synthetic polymer composite. Besides, the typical iodine content in PVP-I is 10 wt.%, which indicates that the influence of PVP in the composite should be considered.

Herein, an orthogonal design approach was used to achieve the optimal experiment conditions in UiO-66-NH_2_ adsorbing iodine. Then, iodine immobilized UiO-66-NH_2_ (UiO66@I_2_) nanoparticles were added to the PCL matrix with a low weight percentage. The PCL composite exhibited effective antibacterial performance. Due to the low content, MOFs have an ignorable effect on the thermal and structural properties of PCL, except for a decrease of the contact angle. Compared with the free iodine/PCL composite, the addition of MOFs enhanced the antibacterial effect and reduced the loss of iodine in the PCL, suggesting the potential applications in antibacterial fields.

## 2. Materials and Methods

### 2.1. Materials

PCL (M_n_ = 80,000) was purchased from Sigma-Aldrich (Sigma-Aldrich China, Beijing, China) and used without further treatment. ZrCl_4_, 2-aminoterephthalic acid, acetic acid, dimethylformamide, dichloromethane, acetone, ethanol, and iodine were purchased from Sinopharm Chemical Reagent Co., Ltd. (Shanghai, China). Slide cover glasses (Aladdin Biochemical Technology Co., Ltd., Shanghai, China) were cleaned in acetone, ethanol, and DI water in an ultrasonic bath (Kunshan Ultrasonic Instrument Company, Soochow, China) sequentially.

### 2.2. Synthesis of UiO-66-NH_2_ and Adsorbing Iodine

The synthesis of UiO-66-NH_2_ nanoparticles was similar to our previous report [19]. The prepared UiO-66-NH_2_ nanoparticles were collected by a vacuum freeze-drying method. Iodine was dissolved in a potassium iodide solution to form a homogeneous solution (50–200 mg/mL). Then, various amounts of UiO-66-NH_2_ nanoparticles (the mole ratio of iodine to MOF ranged from 50–200) were added to the previous solution under magnetic stirring. The mixed solution was embedded in an aluminum foil to avoid the light and placed into a water bath under different temperatures (20–60 °C) lasting various times (4–24 h). Finally, the UiO66@I_2_ nanocomposites were collected by a centrifugation method and dried by freeze-drying.

### 2.3. Orthogonal Experiments

To optimize the iodine loading capacity, an orthogonal experiment was applied [20]. Iodine in potassium iodide concentration (A), iodine to UiO-66-NH_2_ mass ratio (B), reaction time (C), and reaction temperature (D) were set as dependent variables. The iodine loading percentage (IL) was set as a dependent variable. The iodine loading formulation was arranged according to a four-factor, three-level orthogonal table [L_9_(3^4^)] (Table 1).

### 2.4. Preparation of UiO66@I_2_/PCL Composite

The preparation of the composite was according to a previous report with a minor modification [10]: 1.1 mg of UiO66@I_2_ was dispersed into 400 μL of dichloromethane to form suspension A and 40 mg of PCL was dissolved in 200 μL of dichloromethane to give solution B. Suspension A was sonicated for 30 min before being added to solution B. The mixture was further sonicated for 30 min and then cased on the clean slide cover glasses. The solution was placed in an oven at 37 °C for 3 h to evaporate the solvent. The composites with iodine content 0.5 wt% and 1.0 wt.% were named as UiO66@I_2_/PCL 0.5% and UiO66@I_2_/PCL 1.0%, respectively. Pure and free iodine/PCL composites were prepared in the same procedure as described above. The polymers with iodine content of 0.5 wt.% and 1.0 wt.% were labeled as I_2_/PCL 0.5% and I_2_/PCL 1.0%.

### 2.5. Determination of Iodine in UiO66@I_2_ and UiO66@I_2_/PCL Composite

The content of iodine in the composite was determined by potentiometric titration (ZD-2 automatic potential titrator, Shanghai INESA Scientific Instrument Co., Ltd., Shanghai, China). For the UiO66@I_2_ sample, the supernatant was used, while for the UiO66@I_2_@PCL composite, the sample was firstly dissolved in dichloromethane and extracted to the potassium iodide solution. The titration principle was based on the Equation (1) using the potentiometric titrator. The potential jump point indicated the end point of the redox titration, and the effective iodine content in the sample was calculated according to the Equation (2).
(1)$$I_{2}+2S_{2}O_{3}^{2-}=S_{4}O_{6}^{2-}+2I^{-}$$
(2)$$m\left(I_{2}\right)=\frac{c\left({\mathrm{Na}}_{2}S_{3}O_{3}\right)\times V\left({\mathrm{Na}}_{2}S_{3}O_{3}\right)\times 253.8\times 10^{-3}}{2}$$
where c(Na_2_S_2_O_3_) is the concentraion of sodium thiosulfate titrant (mmol/L), V(Na_2_S_2_O_3_) is the consumed volume (mL), 253.8 is the relative molecular mass of iodine (g/mol), and the calaulated mass of iodine is mg.

### 2.6. Antibacterial Activity Measurement

The zone of inhibition test (the Kirby–Bauer test) was used in determining antibacterial activity of the composite. *Escherichia coli* (*E. coli*, ATCC 8739) was cultured in LB medium under shaking at 37 °C for 10 h. Then, bacteria were collected by a centrifugation method and washed with saline solution one time. The harvested *E. coli* cells (around 10^9^ CFU mL^−1^) were suspended in a fresh medium and spread over an agar plate using a sterile swab. Then, the plates were incubated in the presence of different composites for 24 h. The performance of antibacterial activity was judged by the diameter of the zone of inhibition. The zone of inhibition test for *Staphylococcus aureus* (*S. aureus*, ATCC 25323) was the same as the previous process.

### 2.7. Characterization

Field emission scanning electron microscopy (SEM, TESCAN, MALA3 LMH) was used to observe the morphology of MOFs and composite. UV-Vis absorption spectra were collected using a UNIC UV2600 spectrometer. XRD 6100 (SHIMADZU) diffractometer with Cu Kα irradiation, generated at 30 kV and 10 mA, was used to determine the crystalline structure of the samples. Thermogravimetric analysis (TGA) was performed on a STA 449F5 under a nitrogen atmosphere at the rate of 10 °C min^−1^ from room temperature to 800 °C. The FT-IR spectrum was recorded by a Nicolet iS50 Fourier transform infrared spectrometer. Static water contact angles were measured with POWEREACH^®®^ JC2000D1 (Shanghai Zhongchen Digital Technic Apparatus Co., Ltd., Shanghai, China) at room temperature, with ultrapure water chosen as the probe liquid.

## 3. Results and Discussion

### 3.1. UiO-66-NH_2_ Loading Iodine

The octahedral MOF nanoparticles with an average size around 170 nm were shown in Figure 1. The crystallinity was characterized by powder X-ray diffraction (PXRD) with three distinctive peaks at 2θ = 7.38, 8.56, and 25.67°, indicating UiO-66-NH_2_ was successfully synthesized [21]. The iodine loading capacity was optimized through an orthogonal experimental design (Table 1). The orthogonal design is the main method of the fractional factorial design. When three or more factors are involved, the orthogonal design achieves an equivalent result to a large number of comprehensive tests with a minimum number of tests. The results of optimization of the iodine loading capacity were listed in Table 2. The iodine loading percentage (IL) ranged from 5.64 wt.% to 18.45 wt.%. The R values represent the significant effect of the factors. The higher the R values are, the more significant the factors are [22]. Therefore, the effect of four factors decreased in the following order: A > D > B > C, indicating A, i.e., the concentration of iodine in potassium iodide solution, was the most important one among the factors. In Table 2, K1, K2, and K3 represented the mean values of the evaluation index for the three levels corresponding to one factor [23]. The higher concentration of iodine would contribute to the iodine loading. Therefore, the *K* value should be chosen to be as large as possible. As a result, the optimal level combination of factors was A_1_B_2_C_2_D_2_.

### 3.2. Preparation and Characterization of UiO66@I_2_/PCL Composite

To prepare UiO66@I_2_/PCL, UiO66@I_2_ with 18 wt.% iodine was used due to the high loading capacity. The nanoparticles were freeze-dried and dispersed in dichloromethane (DCM) by an ultrasonic vibration to obtain a homogeneous suspension. The nanoparticles’ suspension was added to the PCL/DCM solution ultrasonically before being cast to the glass slide. The sample was obtained by evaporating the DCM at 40 °C in an oven. After evaporating the solvent, the appearance of the UiO66@I_2_/PCL film changed from white to yellow, the same to its solution color. However, the appearance of the I_2_/PCL film was white, although its solution was yellow. The morphology of the composite films under top view was observed by SEM, as shown in Figure 2. The pure PCL membrane gave a net surface, which was consistent with the I_2_/PCL membrane. After adding UiO66@I_2_, micro-sized nanoparticles (red circle in dashed line) were found, indicating the dispersion of nanoparticles was not even, as expected. The ultrasound method could help disperse the nanoparticles in the solution. However, the high viscosity of the PCL solution and high surface energy of the nanoparticles induced the agglomerations. The defects from the open Zr sites or modulators could improve the adhesion on the interface of the PCL to Zr-MOFs by coordinating interactions between the Zr sites of the MOFs and the carbonyl groups of the PCL [10]. Thus, even the high content, such as 50 wt.% of UiO-66 in the PCL, could disperse well without agglomerations. Similarly, the surface of Zr-MOF was occupied by the iodine molecules, which would be caused by the high surface energy and the coordination bonds between Zr and iodine. Therefore, there was no free Zr site on the surface, which resulted in the aggregation of nanoparticles in the PCL matrix. In the FTIR spectra, the peaks at 2950 cm^−1^ and 1720 cm^−1^ were the C-C bond and C=O bonds from the PCL (Figure 2d). The presence of MOFs was confirmed by the XRD pattern, as shown in Figure 2e. The crystalline peaks at 22.01 and 24.17 were attributed to the (110) and (200) of the PCL. Compared with pure PCL, new peaks at 7.38°, 8.56°, and 25.67° in the UiO66@I_2_/PCL membrane belonged to Zr-MOFs. A static water contact angle was used to evaluate the surface wettability of the membrane. The pure PCL had a hydrophobic surface with a contact angle around 93°. After doping the UiO66@I_2_ nanoparticles, the contact angle decreased to 83° and 82° for the 0.5 wt.% and 1.0 wt.% addition, respectively. In contrast, if doping with iodine, the contact angle did not change. Obviously, the hydrophilic surface is easier for cell adhesion and growth, benefiting their biomedical applications.

### 3.3. Antibacterial Properties of UiO66@I_2_/PCL

The zone of inhibition test was applied to evaluate the antibacterial activity of PCL, UiO66@I_2_/PCL, and I_2_/PCL. As is known, UiO-66-NH_2_ and UiO-66-NH_2_/PCL have no antimicrobial effect and iodine is a commonly used biocide. Therefore, the antibacterial activity of the composite membrane is determined by the iodine. It was found that UiO66@I_2_/PCLs had an effective antibacterial activity towards both *S. aureus* and *E. coli*, which represented the *Gram*(+) and *Gram*(−) bacteria. In contrast, I_2_/PCLs with the same iodine content had no antimicrobial activity. The diameter of the inhibition zone was used in the quantification of antibacterial activity. The diameters of the inhibition zone tested by different PCL composites are summarized in Figure 3b. Obviously, UiO66@I_2_/PCL 1.0% had a larger inhibition zone diameter than UiO66@I_2_/PCL 0.5%, indicating that the higher content of UiO66@I_2_ lead to the higher antibacterial activity. Compared with *S. aureus*, the diameters of the inhibition zone by the UiO66@I_2_/PCL membrane against *E. coli* were much larger, which suggested the high antibacterial performance against *E. coli*.

As shown in Figure 3a, the same prepared iodine content in UiO66@I_2_/PCL and I_2_/PCL composites resulted in different antibacterial activity. Due to easy evaporation and sublimation, the real iodine content in the composite could be different than their theoretical content. Thus, we dissolved the polymer composite in DCM, extracted iodine to a potassium iodide solution, and determined the iodine content by the sodium sulfite titration method. As shown in Figure 3c, the real iodine contents in the UiO66@I_2_/PCL and I_2_/PCL were varied. The iodine content in PCL was decreased to only 0.04 wt.% and 0.06 wt.% for I_2_/PCL 0.5% and I_2_/PCL 0.5%, respectively. As iodine was the active ingredient in the antibacterial composite, therefore the I_2_/PCL composites exhibited a negative effect on antibacterial performance. Free iodine is a physiologically active molecule, which is easy to evaporate and sublime during the preparation and storage. The high surface area and open Zr site of UiO-66-NH_2_ provide strong affinity towards free iodine, which supports the high iodine loading capacity and decreases the volatile property. During the preparation process, the loss of iodine is hindered by MOF. Thus, there was a small decrease in iodine content for the UiO66@I_2_/PCL composite (Figure 3c). The iodine content in the composite was stable for 1 week under the light. Polyvidone-iodine (PVP-I) is a common additive in polymers to endow the antibacterial activity [24,25,26,27]. The iodine in PVP-I is usually 10 wt.%, which means most of the addition is PVP. Additionally, PVP is an amorphous polymer that could decrease the physical properties of the composite. In contrast, no similar problems exist in UiO66@I_2_ due to the high iodine loading capacity and crystalline structure of MOF.

## 4. Conclusions

In conclusion, UiO-66-NH_2_ nanoparticles could load iodine as high as 18 wt.%. The iodine immobilized UiO-66-NH_2_ (UiO66@I_2_) nanoparticles were effective antibacterial additives for PCL towards both *S. aureus* and *E. coli*. Due to the high surface area and open Zr sites, the UiO-66-NH_2_ nanoparticles had a high iodine loading capacity. The MOF nanoparticles in the PCL had some aggregations because of the surface occupied by iodine molecules. UiO66@I_2_/PCL composites had a smaller contact angle than PCL, benefiting their biomedical applications. The composites showed a strong antibacterial activity in a low iodine content. In contrast, no antibacterial activity was found in I_2_/PCL composites. The difference in the antibacterial performance was attributed to the reduced loss and stabilized iodine by MOF. The loading and stabilizing iodine by MOF provide a convenient way for fabricating iodine-based antibacterial nanoparticles, polymers, and composites.

## Figures and Tables

**Figure 1 polymers-14-00283-f001:**
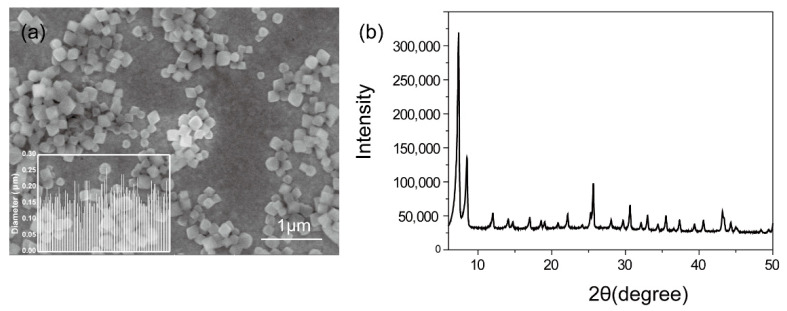
(**a**) SEM images of UiO-66-NH_2_ nanoparticles (insert: histogram graph of diameter). (**b**) XRD pattern of synthesized UiO-66-NH_2_ nanoparticles.

**Figure 2 polymers-14-00283-f002:**
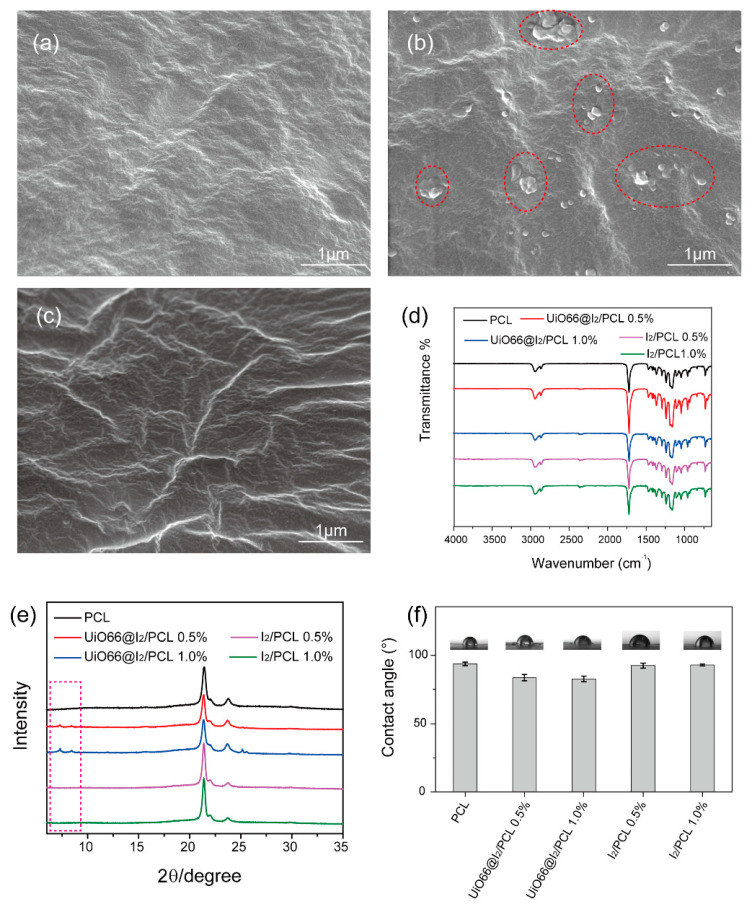
SEM images of (**a**) PCL, (**b**) UiO66@I_2_/PCL, and (**c**) I_2_/PCL composites. (**d**) FTIR spectra of PCL, UiO66@I_2_/PCL, and I_2_/PCL composites. (**e**) XRD pattern of PCL, UiO66@I_2_/PCL, and I_2_/PCL composites. (**f**) Static water contact angles of PCL, UiO66@I_2_/PCL, and I_2_/PCL composites.

**Figure 3 polymers-14-00283-f003:**
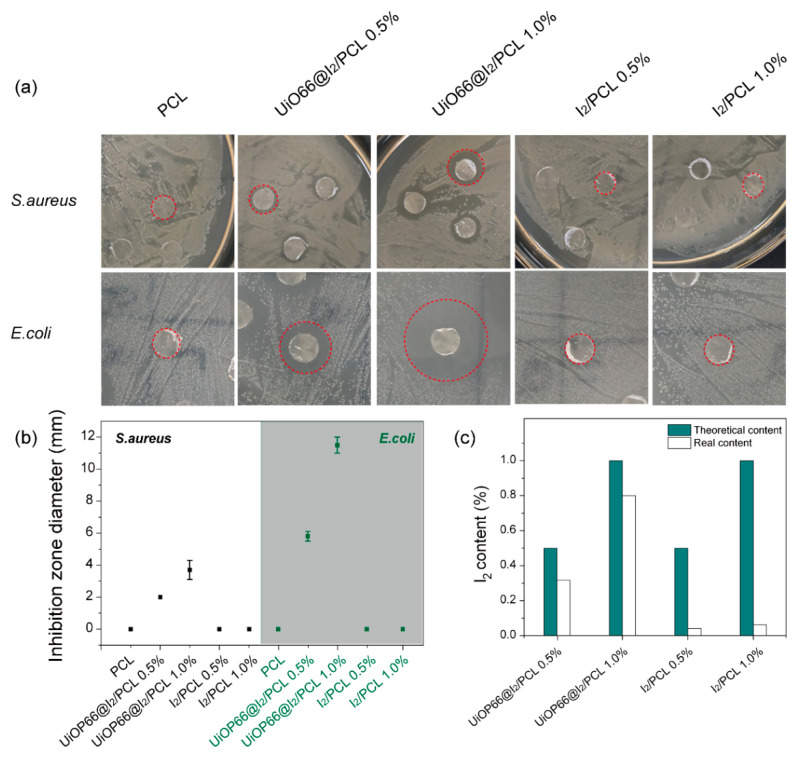
(**a**) Plate photographs of the zone of inhibition test for the PCL composite. (**b**) Inhibition zone diameters corresponding to different composites. (**c**) The theoretical and real iodine contents of various composites.

**Table 1 polymers-14-00283-t001:** Factor-level arrangement table.

Level	A (mg/mL)	B (mol/mol)	C (h)	D (°C)
1	50	50	4	20
2	100	100	8	40
3	200	200	24	60

Note: A: concentration of iodine in potassium iodide solution; B: iodine to UiO-66-NH_2_ mole ratio; C: reaction time; D, reaction temperature.

**Table 2 polymers-14-00283-t002:** Experimental arrangement and results based on the L_9_ (3^4^) orthogonal table.

FN	A	B	C	D	IL(%)
1	1	1	1	1	11.10
2	1	2	2	2	18.45
3	1	3	3	3	12.69
4	2	1	2	3	4.98
5	2	2	3	1	12.69
6	2	3	1	2	10.64
7	3	1	3	2	9.13
8	3	2	1	3	5.64
9	3	3	2	1	12.10
K1	14.08	8.40	9.12	11.96	
K2	9.43	12.26	11.84	12.74	
K3	8.95	11.81	11.50	7.77	
Range	5.13	3.86	2.73	4.97	

## Data Availability

The data presented in this study are available on request from the corresponding author.

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
