# Peer review of "Iodine Immobilized UiO-66-NH2 Metal-Organic Framework as an Effective Antibacterial Additive for Poly(ε-caprolactone)"

_polymers, 2022, doi:10.3390/polym14020283_

Round 1
Reviewer 1 Report
In biomedicine and packaging design, the breakthrough innovations accompanying the transition from synthetic material to degradable bio-based one should save the antibacterial activity of the latter. The submission is devoted to the development of novel antimicrobial nanocomposite on the base of PCL composites embedded by UiO-66 in combination with iodine. The idea of an iodine equivalent decrease as the factor that reduces the structure-morphology disturbance has been successfully executed by the authors at saving the antibacterial agent efficacy.
The composition of the manuscript is logically adequate. In principle, the abstract reflects the general items and the intentions of the authors. The introductory section provides a relatively comprehensive background of the topics that immediately allows the reader to acquaints with the principal trends of the PCL hybrid modifications. The graphical status for all figures and tables is quite acceptable. The text and terminology are transparent and understandable for the experts.
Under the multiple factors, to minimize the number of trials, the authors have used an orthogonal experiment formalization. However, it is worth recommending to create the special section looking as “Orthogonal experiments design” tentatively like in https://doi.org/10.1016/j.ijbiomac.2009.04.020 for disclosing the idea of the experiment treatment
Fig. 3b has undoubtedly demonstrated the benefit of the UiO66 – PCL -iodine composite as an antibacterial material.
UiO-66 – please give the determination of this abbreviation. Is this a Zr-containing compound?
Summing up, I recommend this paper for the following Edition performance after making the above minor amendments.
Best regards in New Year
Reviewer 2 Report
This study aimed to synthesize the UiO-66 loaded iodine/Poly(ε-caprolactone) nanocomposite for effective antibacterial properties. This study have contains no significant results and novelty. For example, iodine is loaded to the UiO-66/PCL composites to improve the antibacterial activity. But contrastingly, the study is reported that the same content of iodine and iodine-loaded PCL composite did not show any antibacterial activity. If so, how composite could show the antibacterial activity, and what is the basic purpose of iodine loaded into the composite. Because of those insignificant results and data correlation, I could not recommend the manuscript for publication at present form.
Some critical suggestion
- The manuscript is written poorly, very hard to understand some sentences. For example, the first two lines of the abstract.
- How authors can prove how UiO-66 nanoparticles reduce the iodine loss and improve stability?
- How the iodine loading is measured?
- Why no differences in the FTIR spectra among the sample?
- Figure 3 and its results are not correlated scientifically.
- The material was poorly characterized and the biological study was also not enough to conclude the results.
Reviewer 3 Report
The manuscript focuses on the preparation of poly (e-caprolactone)-based antibacterial composites with a low antiseptic content such as not to affect the physical properties of the matrix.
In particular, the authors consider the use of iodine-doped metal-organic nanostructures (MOFs) to obtain new materials with an antibacterial concentration (I2) equal to 0.5% and 1.0% by weight.
The new materials were subjected to an in-depth physical analysis with appropriate characterization techniques (antibacterial activity, morphological and UV-FTIR spectrophotometric analyses, crystallinity, thermogravimetric analysis and contact angle).
The results are clearly reported and discussed with the support of an adequate number of figures and tables.
Recognizing the scientific relevance of this study with evident repercussions in sectors with high added value such as biomedical and appreciating the advancement of knowledge on the topics covered, it is believed that this work can be directly accepted for publication.
Round 2
Reviewer 2 Report
-